# Validation of the Attitude Scale on Prospective Teachers’ Perceptions of the Consequences on Their Psychological State: Well-Being and Cognition

**DOI:** 10.3390/ijerph20085439

**Published:** 2023-04-07

**Authors:** Carlos Hervás-Gómez, María Dolores Díaz-Noguera, Ángela Martín-Gutiérrez, Gloria Luisa Morales-Pérez

**Affiliations:** 1Departamento de Didáctica y Organización Educativa, Universidad de Sevilla (US), 41013 Seville, Spain; 2Departamento de Teoría e Historia de la Educación, Universidad Internacional de La Rioja (UNIR), 26006 Logroño, Spain; 3Departamento de Teoría e Historia de la Educación y Pedagogía Social, Universidad de Sevilla (US), 41013 Seville, Spain; 4Centro Adscrito de la Universidad de Sevilla (US), Escuela Universitaria de Osuna, 41640 Osuna, Spain

**Keywords:** collaboration, emerging pedagogies, higher education, motivation, well-being, situated learning

## Abstract

The aim of this study was to analyse the validated psychometric characteristics of the “Scale of Attitudes towards New Post-Pandemic Scenarios” (SANPS) tool using a short version on Perceptions of Future Teachers towards the New Post-Pandemic Educational Scenarios; describe the attitudes of future teachers towards motivation, collaboration, and emerging active pedagogies; and determine the internal consistency and reliability of the tool. The design structure of the instrument consists of the following three latent factors, which were obtained through an exploratory factor analysis (EFA): empowerment/motivation, autonomy/situated learning and emerging digital pedagogies. The questionnaire was administered to a sample of 966 participants. In the confirmatory factor analysis (CFA), a previous hypothesis was established regarding the relationship of the factors and their number and nature, specifying the number of factors and the way in which the variables are related to each other. The 66.53% of total variance was explained. The reliability, calculated with Cronbach’s alpha, reached a global value of over 0.90 (α = 0.94). This valid and reliable questionnaire, which incorporates a dimension that measures the transfer of learning in hybrid and multimodal models of digital ecosystems in Higher Education, can be applied in the evaluation of online education processes.

## 1. Introduction

This section deals with the theoretical contextualisation of the study, as well as those factors that have been taken into account in the validation: motivation, collaboration and emerging active methodologies.

### 1.1. Contextualization

The pandemic experienced worldwide in recent years has left its mark on our lives, and when we talk in education about the side effects caused by the pandemic, we are referring to the positive well-being necessary for students to be able to develop their meaningful learning in the best possible conditions. Subjective well-being, understood as students’ assessment of the satisfaction they derive from the activity, identifying both positive and negative emotions, the conclusions of the latest research related to this psychological factor underline social relationships and the quality of interpersonal relationships as essential keys to meaningful learning. These factors can be equated with health or income level, for example, given their importance [1]. Other research shows how the presence of stress and lack of wellbeing favours the appearance of mental disorders, which have increased after experiencing the implications of the reclusion and loneliness to which these students were subjected during the pandemic. We know how their distress grew as they cancelled and cancelled all their social activities [2]. We also have enough data to reinforce the idea that peer interaction is one of the most significant variables in the teaching and learning processes in Higher Education [3]. Part of our commitment is to offer alternatives to alleviate and help to overcome the learning models that students had to suffer in universities due to the health impositions of the time. In this aspect, we want to focus our research on the archetype of the teaching processes in Higher Education, from the point of view of hybrid teaching, with flexible schedules, and we want to focus on the links established by the students. We must seek the integration of learning models with the social context, and students must develop their cognitive as well as their social and emotional competences [4]. Throughout this paper, we will develop the variables that we consider will favour subjective well-being, namely: motivation, emerging digital pedagogies and situated learning.

### 1.2. Motivation

Motivation is the internal state we seek to activate in our students. For many decades, researchers in education have explored the concept of motivation, because of its dynamic character, meaning movement or motor of human activity. It is for this reason that many of the working hypotheses in research studies have been focused on identifying the internal state of students, with the implication of making an X-ray that tells us what tasks or mechanisms we should develop to achieve activation in learning [5].

Other elements that are closely linked to achieving this activation of learning have been, for example, the experiences that have been produced in abnormal learning situations, as has been the case with the design of new learning scenarios with the pandemic we suffered with COVID-19. In this case we would especially like to emphasise the concepts of autonomy and self-regulated learning. As other studies affirm, motivation was the engine that managed to make up for the lack of preparation of educational institutions for a scenario such as the one experienced during the pandemic [6]. These are key elements of the New Post-pandemic Scenarios [7], and the development, autonomy and self-regulated learning of our students is a fundamental challenge. These are key elements of the New Post-Pandemic Scenarios, and as some authors affirm, the development, autonomy and self-regulated learning of our students has been and will be a fundamental challenge [7].

Overcoming this challenge will allow students to incorporate the new social projection with emerging working environments under continuous management of crises (health, economic or social), and thus they must be very sensitive to the demands of development agendas such as the 2030 Agenda [8]. According to different authors who have written about the generation of knowledge, learning is considered ubiquitous [9], invisible [10], connected [11] and rhizomatic [12], as new formats are incorporated and times and spaces are expanded or modified. Future teachers will have to respond to a set of emerging capacities and skills. This entails that taking the initiative in their own working life and responsibility for their own future is decreasingly based on the past. This is not a crisis, but a change of times. We are in the Age of Innovation. To understand the nature of this new age, it is only necessary to check the literature regarding the impact of change [13,14]. We will have to reinvent education and the concept of globalisation, because the digital transformation will modify the educational scenarios. What is required for this transformation? The digital transformation of an organisation requires incorporating technologies, creating or modifying processes and having the people with adequate capacities and skills for such processes and technologies.

### 1.3. Emerging Digital Pedagogies

Educational communities are made up of users who share similar views and behaviours and constitute knowledge networks. Emerging digital pedagogies refer to new pedagogical approaches and practices that emerge in response to technological advances and the increasing presence of digital technologies in education. These pedagogies focus on the effective integration of digital technologies in teaching and learning, with the aim of improving the quality of education and the learning experience of students. The emerging digital pedagogies include the following: Digital project-based learning, which engages students in creating digital projects to demonstrate their understanding of subject concepts and topics; Online and distance learning, which allows students to access learning materials and communicate with teachers and classmates online; Personalised learning, which uses technology to tailor the learning experience to the needs and preferences of individual students; Gamification, which uses gaming elements to motivate students and encourage participation and engagement in the learning process; and Mobile learning, which allows students to access learning materials and perform learning activities on their mobile devices, such as smartphones and tablets. In general, emerging digital pedagogies aim to harness the potential of digital technologies to improve the quality and effectiveness of the teaching and learning process, and to prepare students for the ever-changing digital world [15,16].

Therefore, the changes in the interactive learning environments are ecologically related between individuals and communities. In this sense, studies have been focused on academic performance, learning efficacy (both cognitive and emotional), satisfaction and self-efficacy. Baturay (2011) identified a strong relationship between the proposed content and students in interactive learning [17]. Consequently, it is especially interesting to know how the learning ecologies and the communities that compose them adapt and adjust to the changes in the teaching environment caused by the COVID-19 pandemic. It is particularly relevant to observe in higher education and in the training of future teachers. Understanding the significant consequences of the pandemic and its impact on education, it is a priority to incorporate digital literacy processes. This implies the use of technology-mediated pedagogical strategies, as this favours cognitive development and improves teaching practice [18]. Thus, study must go beyond isolated actions of digital education. Universities must design and develop adequate strategies, for instance, in the case of teacher digital competence, with policies of teacher recognition, counselling and support in the use of digital resources [19]. Similarly, another study has pointed out the key elements of institutional strategies [20] and other work highlight how emergency pedagogical practices can be an opportunity to review models, create new conditions and imagine a transformed school [21].

The aim of the teacher is to ensure that the students develop critical thinking, analyse the content in the overflow of information and integrate the emerging digital resources and tools through the execution of activities that allow them to express themselves, creating digital products and carrying out communication processes collaboratively with their classmates in networks.

### 1.4. Situated Learning

Situated learning is a theory of learning that emphasises the importance of learning in authentic and relevant contexts where knowledge has application. Situated learning is based on the idea that knowledge and skills are not acquired simply through direct instruction but are constructed and internalised through active participation in authentic and meaningful situations. Thus, situated learning emphasises practical learning and experience as a way of learning and applying knowledge. It is for this reason that the socio-cultural context is a key element in the acquisition of skills and competences, and as learners achieve these they seek and find solutions to the challenges we face, albeit with a collective vision.

Situated learning is learning that depends on the activity, the context and the culture in which it takes place. It needs four phases to take place: connection with reality, reflection, collaboration and transfer.

This methodology favours collaborative work styles and improves relationships between project members. The theory of situated learning was proposed by [22] in the context of communities of practice. Situated learning seeks to encourage team and cooperative work through problem-oriented projects that require the application of analytical methods that take into account all kinds of relationships and linkages. We are taking part in the great challenges that this century has brought us; interdependence is now valued as necessary when we are objects of conflict, and in this sense the theory of social interdependence [23] induces us to underline the importance of cooperative learning. Above all other considerations, situated learning tries to reflect the importance that the social dimension is currently acquiring in the construction of knowledge, as well as the value of meaningful knowledge. For this reason, the aim of this teaching method is not only to be limited to classroom learning, but also to adapt to virtual learning (e-Learning, LMS platforms) and to the work environment.

## 2. Methodology

In this methodology section, we find the objectives, the sample (participants, procedure) and data collection and analysis.

### 2.1. Objectives

The aim of this study was to analyse the validated psychometric characteristics of a short version of the “Scale of Attitudes towards New Post-Pandemic Educational Scenarios” (SANPS) concerning the Perceptions of Future Teachers. The tool was designed by researchers from a Spanish university with the purpose of diagnosing the new educational scenarios that arose after the COVID-19 pandemic. The study focused on describing the attitudes of future teachers towards motivation, collaboration and emerging active pedagogies, as well as determining the internal consistency and reliability of the aforementioned tool. The shortened version of the “Scale of Attitudes towards New Post-Pandemic Educational Scenarios” (SANPS) was utilized in order to obtain a reduced version that would allow for the necessary agility for the proposed scenarios, establishing the design and validation of an instrument that allows measuring the attitudes of future teachers towards digital education in post-pandemic scenarios. By reducing the number of items, a quicker response could be provided, thus making the participating individuals offer a more precise answer and be more active when responding. In any case, no short version of the questionnaire has been found published that could make it possible to quantify the perceptions and attitudes of future teachers regarding post-pandemic educational scenarios.

### 2.2. Sample and Data Collection

#### 2.2.1. Participants

To carry out the present study, different participant profiles were included, attending to the three time points of the research process:Participants in the expert validation: A total of 30 professionals were selected, of whom 13 were experts in digital education/training (43.34%), 12 were experts in measurement instruments design and validation (40%), and 5 were members of the research team (16.66%). The participants were selected by non-probabilistic judgment sampling [24] or purposive sampling [25], in order to guarantee the collection of relevant information regarding those directly involved in the research topic. For the sample selection, a specific group was made with people who could complete the questionnaire, among whom we selected people with extensive experience in the teaching–learning process mediated by technology and the education of future teachers, because they have greater knowledge of the research topic.Participants in the factor analysis and in the reliability of the instrument: The sample was selected by non-random sampling, accepting 5% margin of error and 95% confidence level. Specifically, a non-probabilistic and causal sampling was performed, in which the most common selection criteria were based on the accessibility of the participants, because the request for participation was sent to all the people registered in degrees related to Education in the Spanish University. Thus, a total sample of 966 students of the Spanish University completed the questionnaire, with 696 women (72%) and 270 men (28%) aged 18–25 years (mean = 20.7 years), who were registered in the Faculty of Education Sciences of the Spanish University in the academic year 2021–2022. These values were above those calculated for the confidence level.

#### 2.2.2. Procedure

This work was conducted in three phases: (1) validation of the instrument by experts in the topic and in research methods; (2) determination of the construct validity after passing the version of the questionnaire produced in the previous phase; (3) calculation of the instrument reliability; and (4) validation of the proposed model, specifying the number of factors and the way in which the variables were related. After these phases, the questionnaire was designed and validated for its final version.

In this way, and following the rules established by the European Commission for the development of questionnaires [26], the initial phase was carried out in the following manner: (1) literature review; (2) specification of the study objectives; (3) conceptualisation and operationalisation; (4) exploitation of concepts; (5) definition of variables and planning of tables; (6) decision on the information gathering method; (7) writing and sequencing of the questions; and (8) elements of the virtual design.

For the development of the instrument, the Motivated Strategies for Learning Questionnaire [27] was used to assess motivation towards online learning, and the scale developed by Liu et al. (2011) for the analysis of digital transformation [28]. This instrument consisted of 37 items (5 items on identification and 32 items on digital transformation), which were grouped into five categories: learner profiles, resources (hardware-software), professional collaboration, digital pedagogy and learner motivation. The learner profiles included demographic questions: (1) gender, (2) age, (3) year of degree, (4) group and (5) degree. The remaining questions had a Likert-type scale from 1 (strongly disagree/slightly agree) to 5 (strongly agree/strongly agree). The new items were recoded before analysis (items 9, 21, 22 and 28).

After this first work, the group of experts was selected to perform the validation of the questionnaire, with the 30 professionals mentioned in the participant section. They were given the guide for the content validation of the instrument, which they completed, attending to its relevance, clarity and adequacy. Kendall’s test was applied to assess the agreement between the evaluators.

Once this phase of the procedure was completed, the questionnaire was revised, designing a new version for its final validation. The instrument was analysed, studying its structure through an exploratory factor analysis using the factor extraction method of principal components and promax rotation, because there was correlation between the dimensions, thereby requiring the extraction of factors that are detailed in further sections. To this end, the Kaiser–Meyer–Olkin (KMO) test for sampling adequacy and Barlett’s sphericity test were applied. The sample of this study consisted of a total of 966 students of the Spanish University (Mentioned in the participants section). An online questionnaire was administered to this sample, using the Google Forms platform. This procedure provides simple, fast and inexpensive access to a large number of participants, allowing them to complete the questionnaire in a flexible manner; moreover, this online tool has numerous advantages, such as the direct exploitation of the answers in different formats for their analysis.

The obtained factors and the complete scale were subjected to a procedure of reliability analysis using Cronbach’s alpha and McDonald’s omega to assess the internal consistency of the scale.

Lastly, the construct validity was analysed by confirmatory factor analysis (CFA) to validate the instrument that aims to measure the attitudes of future teachers, and which is defined in the set of attitudes and predisposition of future teachers toward the teaching–learning processes mediated by technology.

#### 2.2.3. Data Analysis

SPSS v.26 software was used for these statistical analyses.

## 3. Results

Because this is a complementary and sequential work, we present the results obtained in each of the commented phases.

Phase 1. Expert validation of the questionnaire

In this first phase, the questionnaire was validated by experts. Table 1 presents the values obtained by items according to the degree of relevance, clarity and adequacy, as was explained in the corresponding section. As can be observed, the mean scores obtained indicate the high degree of agreement between the different experts, obtaining values above 4 out of 5 points in all items, except for item 10, which obtained a mean score of $\overset{–}{x}$ = 3.33 and δ = 1.527 in the section related to the clarity of the item formulation. To conclude the inter-judge analysis, Kendall’s test was applied to evaluate the agreement between evaluators, which was statistically significant (*p* < 0.001; sig. = 0.05).

Phase 2. Exploratory factor analysis

The construct validity of the questionnaire was assessed by exploratory factor analysis using the principal components method, because the analysed study variables are quantitative [29], and this method allows replacing the original variables of the questionnaire with a small number of lineal and uncorrelated combinations, thereby losing little information [30]. The questionnaire was validated, attending to 36 items, because item 10 was eliminated.

In the Kaiser–Meyer–Olkin test for sampling adequacy (KMO), a value of 0.94 was obtained, exceeding the recommended values of 0.60. In Barlett’s sphericity test, a value of 0.0001 (*p* < 0.01) was obtained. These two values are acceptable, indicating that the correlation matrix is not an identity matrix, thus it is appropriate to carry out the factor analysis.

Thus, in the parallel analysis, when comparing the values, the factors whose real eigenvalues exceeded the randomly ordered eigenvalues were retained [31].

Once the principal component analysis was conducted, the three required factors were extracted, which obtained values above 1 (9.19, 1.76 and 1.01), with a total explained variance of 66.53% in the whole scale (51.10% in the first factor, 9.78% in the second factor and 5.64% in the third factor). According to Williams et al. (2010) parallel analysis appears to be among the best methods for deciding how many factors to extract or retain.

As can be observed in the matrix of rotated components (promax with Kaiser normalisation), the first dimension “empowerment/motivation” includes four items; the second dimension “autonomy/situated Learning” includes three items, and the third dimension “emerging digital pedagogies” includes three items (Table 2).

Phase 3. Instrument reliability

This model, obtained in the exploratory factor analysis, was subjected to a reliability analysis. In this case, the procedure of internal consistency was used, as it determines the reliability of the scores obtained through a single administration of the test, generalises the scores with respect to a dimension or set of items and observes whether the participants respond consistently throughout the set of items used [32]. Within this procedure, the correlations between the different parts of the test were considered, taking into account the universe of items. Cronbach’s alpha was used, which calculates reliability through statistics that cannot be applied to nominal variables in which the properties of the numbers cannot be managed, such as order and size [33].

The Cronbach’s alpha obtained for the entire scale was 0.94. In the reliability analysis of each of the factors, a value of 0.87 was obtained for the first factor (empowerment/motivation), 0.90 for the second factor (autonomy/situated learning) and 0.91 for the third factor (emerging digital pedagogies). In general terms, and following [34] (p. 189), the “correlations between 0.8 and 1 are considered as very strong and, consequently, indicate high levels of instrument reliability”; therefore, it could be asserted that the questionnaire presents high levels of reliability, because the lowest factor presents a reliability of 0.87.

The calculation of McDonald’s [35] omega coefficient (1999) provides a more stable value, because it is obtained from the factor loadings, that is, from the weighted sum of standardised variables. The values of both coefficients are gathered in Table 3 and allow asserting that the obtained values are acceptable and, thus, the scale is reliable.

Phase 4. Confirmatory factor analysis

To complete the exploratory factor analysis conducted in previous phases, a confirmatory factor analysis was performed. The aim of this multivariate statistical technique is to evaluate the proposed structure for a data matrix and determine the following: (1) the relationship between the observable indicators and the established constructs, (2) whether the hypothesis proposed with the suggested model is valid and (3) whether the inferences are correct [36].

To perform the CFA, we established a previous hypothesis regarding the relationship of the factors, and their number and nature, which specifies the number of factors and the way in which the variables are related to each other.

According to the theoretical model proposed for the development of this instrument, three dimensions were established. The first dimension “empowerment/motivation” includes four items, the second dimension “autonomy/situated Learning” includes three items, and the third dimension “emerging digital pedagogies” includes three items (Table 4).

After accepting that the sample is large enough to carry out this measurement, and the required multivariate normality [37], the CFA was performed using AMOS SPSS v.26 from a database developed in SPSS v.26 [38].

Figure 1 shows the representation of the relationships between the variables and the factors through a PAD diagram.

According to the structure of the instrument and the proposed hypothesis, there is a principal variable called “attitudes of future teachers”, which is a latent variable, and it is constituted by three dimensions, which are also latent variables because they cannot be measured directly. After carrying out the analysis, the following results were obtained.

As can be observed in the obtained results in Table 5, the Chi-squared value was below 0.05, thus, in principle, it can be considered that it indicates differences between the data matrix observed and the matrix estimated by the model. Although this value could suggest that the proposed model is inadequate, it is relevant to point out that this measure can be affected by the sample size [39], because the larger the sample, the greater the possibility of considering the established model to be inappropriate, thereby making it necessary to assess the rest of the measures of the analysis, such as the root mean square error of approximation (RMSEA). This measure expresses the amount of variability that cannot be explained by the factor model by degrees of freedom. When the sample is large enough, as in this case, the value of RMSEA suppresses the problem posed by the Chi-square likelihood ratio. Table 5 shows that RMSEA is above 0.05, which is the acceptable threshold for this value [39]. 

Regarding the incremental fit measures, the comparative fit index (CFI), the Tucker–Lewis index (TLI) and the normative fit index (NFI) obtained values above 0.9, thus they exceed the threshold of acceptance, i.e., they are acceptable values [39].

Lastly, the parsimony-adjusted measures are also considered acceptable, because PRATIO, PCFI and PNFI obtained values above the desirable threshold of 0.9, and the Akaike information criterion (AIC) obtained a value that was considered too low to be accepted.

Considering all these measures, enough evidence was obtained to accept the proposed factor model, thus it fits the observed data and does not require modifications for adjustment, because the relationships considered previously were not significant. This confirms the construct validity of the instrument based on the CFA technique.

## 4. Discussion

In view of the obtained results, we reviewed the factors presented in a previous study [40]. The aim of this study was to analyse the validated psychometric characteristics of a short version of the Attitude Scale regarding the perceptions of future teachers toward the New Post-Pandemic Educational Scenarios (SANPS): to describe the attitudes of future teachers toward motivation, collaboration and emerging active pedagogies; and to determine the internal consistency and reliability of the Attitude Scale regarding New Post-pandemic Scenarios (SANPS). Thus, the factors that favour the adaptation of students, as we have already seen in other studies, are motivational strategies, fundamentally because motivation is considered as an amalgam of cognitive, metacognitive, motivational and social factors that affect students’ performance and all aspects inherent to the learning process. We also had the opportunity to see how professional collaboration fostered what we have called open pedagogy: content curation, peer feedback, community feeling, participation, conceptual development, reflection and cognitive scaffolding. Therefore, in the section on student empowerment, we were able to identify problem-solving strategies and understandings of situated learning, presenting in the vast majority of cases. Following the model that we have used in previous writings, we can continue to affirm that the factors collaboration, motivation, situated learning and the involvement of active and digital methodologies have determined the ability of students to face the adaptation challenge caused by the COVID-19 pandemic, because practically all of those consulted stand out as determining factors. The results indicate that the non-presence has influenced the effective learning of the students, who felt the need to interact verbally with the teaching staff at least once a week, and this is indicated by the vast majority of the participants; undoubtedly, this suggests the need to improve the online pedagogical model [7].

The attitudes observed invite us to rethink the educational project by integrating the contributions of the students, not as automatons but as actors who make valuable contributions to the construction of their learning. Showing and communicating their own identity must be part of the process [41], and should encourage rethinking the idea of expanded schools that allow the configuration of hybrid models that include technologies in which the modes of construction of learning are problematized and educational experiences are enabled under collaborative modalities and more significant and relevant experiences [21]. In this case, we present a model that relates three factors for digital adaptation in university teaching: autonomy and situated learning, motivation, and emergent, digital and active pedagogies. The experience in the hybrid multimodal method [42] provided us with relevant information on the development of didactic material by students [11]. However, we must continue to analyse our data and the contexts in which we have generated them, given that the determination of favourable or unfavourable contexts is not yet well determined. It is important to advance digital methodologies that allow different spaces and times, where all options are included: face-to-face classes, learning environments that allow asynchronous teaching educational performances, where students can exchange information and interact, do their homework, encourage discussions and have access to all relevant information of the discipline [43]. Finally, we re-emphasise that institutions need to take action and break down digital barriers for students, which will undoubtedly mean progress in equal opportunities and student achievement.

## 5. Conclusions

Considering the perceptions of the students, and attaining the objectives set in the present study, it can be concluded that motivation, collaboration and emerging digital pedagogies are key in the setting of the new learning environments or new post-pandemic scenarios.

In this sense, further studies should delve into the design of working guidelines of autonomous work for higher education students and advance didactic resources, digital pedagogies and, fundamentally, the concept of evaluation [44]. We conclude that factors such as university student autonomy greatly improve learning performance and involve a certain level of adaptability to the new requirements of digital educational transformation. Moreover, due to the importance of the learning design, digitally emerging and innovating pedagogy is a phenomenon that promotes our incorporation as professionals of the 21st century to the new educational age.

The introduction of technological tools is not the only factor necessary to transform teaching–learning activities; the institutions and the actors involved need a transformation that has become more evident in the post-pandemic scenario, bringing evidence of radical changes that still demand changes in communication, of relationships and even roles between those involved, of teamwork, of rethinking responsibility and content creation and even knowledge management or looking further to rethink the educational configuration and content production [45]. The implications for educational institutions are to demand a good integrated design of the teaching process, and design high-performance digital ecosystems, with artificial intelligence, transparent immersive experiences and digital platforms. It is necessary to promote motivation and autonomy in the learning processes of future educators, as well as to create autonomous working environments or digital learning classrooms that allow incorporating a quality education in the new age.

## 6. Recommendations

With this study, the results recommend preferred areas that require attention to improve online education, in particular, student satisfaction [46].

It is essential to reflect on the characteristics of the concepts of “digital learning” [47] and collaborative learning [48], incorporating the design of role-playing games for future studies. Similarly, the relationships and exchange that take place in collaborative learning [48] include suggestions for learning community management. Motivation and learning strategies are composed, as we have had the opportunity to describe, of different scales, which we will further investigate; we do not want to forget cognitive strategies, metacognitive strategies and resource management. For this enterprise, we will go deeper into essay, elaboration, organisation and critical thinking. This structure helps us to continue in the formative process initiated and to advance towards emerging models where most of the digital resources have a place. Similarly, we do not forget the role played by social networks and learning communities [17]. We will continue to design instruments that evaluate new learning scenarios and develop advanced strategies and competences that provide quality learning processes for our students. Therefore, it is urgent to be able to build new approaches that allow the development of intelligent organizations, which in turn allow adaptation to changes in context without losing the ability to provide quality human relationships that allow the development of soft skills such as teamwork and the level of commitment and responsibility necessary for knowledge management [45].

## 7. Limitations

The results of this study were not compared in relation to gender, course or shift, which could imply some differences due to their connotations in terms of context. Moreover, the present work does not provide data regarding another hypothesis that considers the evaluation of the students, the orientation toward learning, and the effort to reach the goals set. It is also important to understand the interests of future teachers in the emerging digital ecosystems. We make some recommendations to organizations based on the attitudes observed: ensure good communication, provide information about the change, engage the students in the making of decisions related to the transformations conducted, adjust the content and teaching method to online learning, take care of social presence using synchronous forms, limit the tools used (preferably choose only one) and offer support in the field of the technologies used, enabling participation in online learning.

## Figures and Tables

**Figure 1 ijerph-20-05439-f001:**
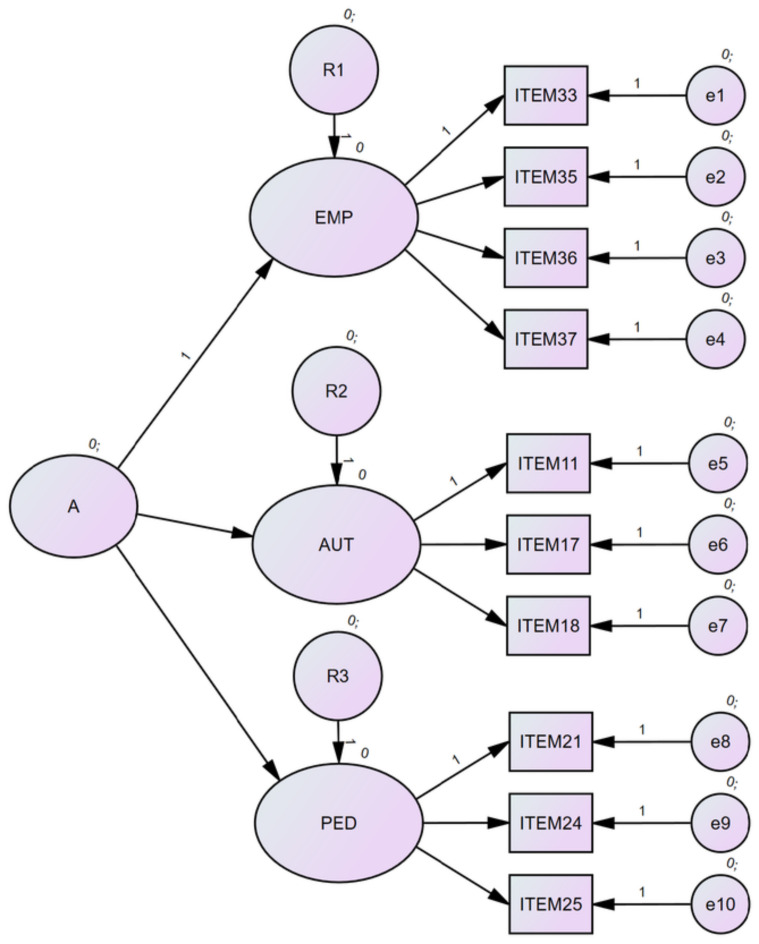
PAD diagram of the causal relationships.

**Table 1 ijerph-20-05439-t001:** Kendall’s test to evaluate the agreement between the evaluators.

	Relevance		Clarity		Adequacy	
Items	$\overset{–}{x}$	δ	$\overset{–}{x}$	δ	$\overset{–}{x}$	δ
Item 1	5.00	0.000	5.00	0.000	5.00	0.000
Item 2	5.00	0.000	4.33	1.154	5.00	0.000
Item 3	5.00	0.000	4.33	1.154	4.33	1.154
Item 4	5.00	0.000	5.00	0.000	5.00	0.000
Item 5	5.00	0.000	5.00	0.000	5.00	0.000
Item 6	5.00	0.000	4.33	0.577	4.67	0.577
Item 7	5.00	0.000	5.00	0.000	5.00	0.000
Item 8	5.00	0.000	5.00	0.000	5.00	0.000
Item 9	5.00	0.000	5.00	0.000	5.00	0.000
Item 10	5.00	0.000	3.33	1.527	5.00	0.000
Item 11	5.00	0.000	5.00	0.000	5.00	0.000
Item 12	5.00	0.000	5.00	0.000	5.00	0.000
Item 13	5.00	0.000	4.33	1.154	4.33	1.154
Item 14	5.00	0.000	4.33	1.154	4.33	1.154
Item 15	5.00	0.000	5.00	0.000	5.00	0.000
Item 16	5.00	0.000	5.00	0.000	5.00	0.000
Item 17	5.00	0.000	5.00	0.000	5.00	0.000
Item 18	5.00	0.000	5.00	0.000	5.00	0.000
Item 19	5.00	0.000	5.00	0.000	5.00	0.000
Item 20	5.00	0.000	5.00	0.000	5.00	0.000
Item 21	5.00	0.000	5.00	0.000	5.00	0.000
Item 22	5.00	0.000	5.00	0.000	5.00	0.000
Item 23	5.00	0.000	5.00	0.000	5.00	0.000
Item 24	5.00	0.000	5.00	0.000	5.00	0.000
Item 25	5.00	0.000	5.00	0.000	5.00	0.000
Item 26	5.00	0.000	5.00	0.000	5.00	0.000
Item 27	5.00	0.000	5.00	0.000	5.00	0.000
Item 28	5.00	0.000	5.00	0.000	5.00	0.000
Item 29	5.00	0.000	5.00	0.000	5.00	0.000
Item 30	5.00	0.000	5.00	0.000	5.00	0.000
Item 31	5.00	0.000	5.00	0.000	5.00	0.000
Item 32	5.00	0.000	5.00	0.000	5.00	0.000
Item 33	5.00	0.000	5.00	0.000	5.00	0.000
Item 34	5.00	0.000	5.00	0.000	5.00	0.000
Item 35	5.00	0.000	5.00	0.000	5.00	0.000
Item 36	5.00	0.000	5.00	0.000	5.00	0.000
Item 37	5.00	0.000	5.00	0.000	5.00	0.000

Source: developed by author.

**Table 2 ijerph-20-05439-t002:** Matrix of rotated components of the exploratory factor analysis model.

Scales	Factor 1	Factor 2	Factor 3
Item 33. I think that learning the contents of this subject is fundamental.	0.824		
Item 35. The contents covered in this subject are of interest to me.	0.908		
Item 36. I am convinced that I can achieve great results in the theoretical and practical part of this subject.	0.801		
Item 37. I consider the resources provided in this subject to be useful for learning.	0.772		
Item 11. The online training affects the relationship with my classmates.		0.899	
Item 17. I think that the learning that takes place in face-to-face sessions is more fruitful than in virtual classes.		0.720	
Item 18. I consider it important to interact face-to-face with the teacher every week.		0.789	
Item 21. Item 22. I think that doing virtual practicals and activities is difficult and requires a period of adaptation.			0.766
Item 24. I consider that virtual education has benefits and is very useful.			0.828
Item 33. I think that learning the contents of this subject is fundamental.			0.841

Source; developed by author.

**Table 3 ijerph-20-05439-t003:** Reliability statistics after the exploratory factor analysis.

	Cronbach’s Alpha	McDonald’s Omega	Nº of Elements
Complete scale	0.941	0.969	10
First factor	0.879	0.905	4
Second factor	0.906	0.948	3
Third factor	0.918	0.930	3

Source: developed by author.

**Table 4 ijerph-20-05439-t004:** Specifications of the future teacher attitudes scale.

Dimension	Items
Empowerment/motivation	33, 35, 36, 37
Autonomy/Situated Learning	11, 17, 18
Emerging digital pedagogies	21, 24, 25

Source: developed by author.

**Table 5 ijerph-20-05439-t005:** Data obtained after the CFA with AMOS SPSS v.26.

	Absolute Fit Measures		Incremental Fit Measures			Parsimony-Adjusted Measures			
Model	Chi-Squared	RMSEA	CFI	TLI	NFI	PRATIO	PCFI	PNFI	AIC
Future teacher attitudes	0.05	0.04	0.94	0.92	0.92	0.96	0.93	0.9	177.14

Source: developed by author.

## Data Availability

Due to confidentiality and privacy agreements, it is not possible to make these data publicly available.

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
