# Peer review of "Validation of the Attitude Scale on Prospective Teachers’ Perceptions of the Consequences on Their Psychological State: Well-Being and Cognition"

_ijerph, 2023, doi:10.3390/ijerph20085439_

Round 1

Reviewer 1 Report (Previous Reviewer 4)

The topic is interesting, but there have been published at least three studies on the same questionnaire (both the short and the long versions). Furthermore, the questionnaire to be validated is problematic. The Discussion section also answers questions that are not supported by the analysis.

Author Response

Thank you very much for your comments. 

On the basis of the reviewer's considerations, the changes are made. 

Reviewer 2 Report (New Reviewer)

What are emerging active methodologies? (line 43)

What are active and emerging digital pedagogies? (line 64) It is unacceptable to use terms without referencing other scholars or offering a definition, examples, etc. While the authors might know digital technologies, I’m not convinced they understand digital pedagogies.

I am concerned about the classification in Table 4. Empowerment and motivation should not be considered as one dimension, the terms are different. Autonomy and collaboration should not be considered as one dimension, the terms are different. The authors need to be specific for the research to be valid. I’m not confident about the trustworthiness of the data.

I instruct pre-service and in-service teachers. I read the article closely two times and struggle as the content is very broad and over-generalized. I have not learned anything new from reviewing this article. I’m excited to learn more about emerging, digital, active pedagogies... Tell me more, and please be specific.

The article needs refinement and careful editing. There is much room for improvement.

Author Response

Thank you very much for your comments. 

On the basis of the reviewer's considerations, the changes are made. 

Reviewer 3 Report (New Reviewer)

Considering the changes and the improvements integrated into the manuscript as well as the content, I suggest the current form of the manuscript be published. 

The references must be checked against the APA style 7th edition (especiallly, date of access issue since it is no longer available in the latest edition of APA style. )

Author Response

Thank you very much for your comments.

Based on the reviewer's considerations, the changes are made.

The standards of the American Chemical Society have been followed. Thank you

This manuscript is a resubmission of an earlier submission. The following is a list of the peer review reports and author responses from that submission.

Round 1

Reviewer 1 Report

Regarding the content, I do not have any changes to recommend, it makes a good literary review to support the relevance of the problem to be studied and a good structuring of the content, it uses the correct methodology for this type of study and it is a consistent and well-detailed methodology to give significance to the results they show, makes a good discussion of the results with respect to the studies carried out previously, and marks the conclusion obtained well.

Although I advise looking at these things:

The point “1. Literature Review” should be called “1. Introduction”, and the point “1.1. Introduction” should be called “1.1. Contextualization”.

Never two sections without a paragraph of text in between. You should put a couple of lines describing/naming the subsections you are going to deal with within that section. You must correct this between sections 1-1.1 and 2-2.1.

And in the section “5. Conclusions”, it is necessary to develop a deeper analysis of the conclusions and implications of the study. In addition to the possible future lines of research opened with this research.

Reviewer 2 Report

This is an article seeking to design and validate a research instrument to measure future teachers' perception towards "New Post-Pandemic Educational Scenarios". The authors utilized various statistical analysis methods (e.g., Cronbach's alpha, Kendall's test, EFA, and CFA ) to determine the reliability and validity of the instrument. However, I think this article suffers from several critical flaws that undermine its credibility.

First, the manuscript lacks a proper introduction that justifies the need and significance for instrument design and validation. It is not very clear what the proposed questionnaire aims to measure. "Teacher's perception" is a very vague construct, and "New Post-Pandemic Educational Scenarios" needs to be better defined.

Second, the literature review should provide a theoretical foundation for selection of key constructs and their measuring items in the instrument. Unfortunately, the literature review is quite brief without connection with the subsequent instrument design.

Third, before implementing the instrument and conducting EFA and CFA, the authors should elaborate on the questionnaire design. For example, how many sub-scales does it have, and what constructs are measured by those scales. What is the rationale for the selection of sub-scales and their inner logic connection... Before validating the instrument using EFA and CFA, first the authors should describe what is the instrument and its theoretical underpinning.

Fourth, it will be more useful to the readers, if the authors can provide an appendix of the validated questionnaire (with revisions highlighted) so that it can be actually used in practice. (Also, it is a good opportunity for authors to think about the fundamental question: Why do we need this instrument in the first place? What can be measured only by this instrument but not the others?

Lastly, I think the discussion section is a bit drifted from the study results. The authors discussed many things that are not relevant to the instrument investigated in this paper.

Reviewer 3 Report

The methodology of the research is carefully described. Each step of the experiment is clear and supported by numerous statistical tools with appropriate connections to the literature. Results have been divided into four phases. In the first phase, the questionnaire was validated by the experts. The procedure for choosing experts is described in the methodology section. The Kendall tau test was applied to evaluate the agreement between experts. The second phase was the exploratory factor analysis using principal component methods. In the third phase, the model undergoes a reliability analysis. The Cronbach’s alpha and the McDonald’s coefficient were calculated. In the last phase, confirmatory factor analysis was performed. The appropriate measures of fit were calculated. The description of the experiment is clear and easy to follow. The statistical tools are correctly chosen. 

I suggest adjusting the layout of the paper in the way that Table 2 is on one page.  In its present form, it is difficult to read the information in column “Factor 3”. In lines 135 and 145 extra spaces between a bullet and the first word should be removed.  

Reviewer 4 Report

The study covers very important and relevant topics. After corrections have been made, it is worth considering publishing the study.

However, it is not clear from the study what the purpose was. For some conclusions it is not clear how they follow from the analysis.

Overall, the context itself is missing to understand the study. Properly contextualised, the conclusions would be useful and valuable.

Round 2

Reviewer 2 Report

I don't think my previous review comments are satisfactorily addressed. The authors referred to reference No. 31 to answer my question regarding questionnaire design, scales and subscales, as well as theoretical underpinning, but reference No.31 is a book on SPSS statistical analysis. Consequently, I am afraid the aforementioned issues still exist in this revised manuscript.

Reviewer 4 Report

Unfortunately, the authors have not responded to the major concerns. A few minor changes have been made, but the more substantial problems have not been addressed.

Even if the number of references is sufficient, the problem of not synthesising previous research has not been addressed. Furthermore, the literature review is meant to summarise what has done before, not to instruct the reader to read it if they are interested ("it is only necessary to check the literature about the impact of change"). 

In the presentation of the purpose of the study, it is still not clear what EANEP is (who created it, on what basis, what is its purpose etc.). They do not write about why a short version of an - unknown - questionnaire is needed.The authors try to give some expanation only in the Discussion, i.e. they copied two sentences from the abstract, which even include the hyphens.

"The aim of this study was to analyse the validated psychometric characteristics of a short version of the Attitude Scale about the Perceptions of future teachers toward the New Post-Pandemic Edu-cational Scenarios (EANEP): to describe the attitudes of future teachers toward motivation, col-laboration and emerging active pedagogies; and to determine the internal consistency and reli-ability of the Attitude Scale about New Post-pandemic Scenarios (EANEP)." 

It is not clear what the exact name of the questionnaire is. There is also a contradiction: the short version is mentioned (short version of EANEP) in the first sentence, but the next sentence refers to the full version (simply EANEP).

The study to which the authors refer (reference 33) gives exactly the same information on the longer questionnaire as this study. The authors of that study also perform the CFA and do not present the longer questionnaire as the authors of this study tell us in their response. That study (ref 33) gets exactly the same factor structure as this study.

So, it still remains unclear what the purpose of this study is.

As for the questions in the questionnaire, the authors write: "The topic addressed is the perception of future teachers". This is their answer to the question what the term "subject" refers to in the items. This answer is peculiar because in this case it is not possible to interpret e.g. item 36 of the questionnaire: "I am sure that I can do a great job in the assingments and exams of this subject.". If the subject is the "perception", how can it be tested in an exam or assignment?

The Discussion section continues to try to answer questions for which no data are provided by the authors.

Overall, the authors' answers are not reassuring and do not answer the main concerns. The answers leave the reviewer uncertain as to whether the authors can provide a meaningful response to the shortcomings found in the first version. On this basis, however, this study is not recommended for publication.
